# Assessing the risk of Guillain-Barré syndrome in older adults after bivalent RSV pre-F vaccination in England

Julia Stowe [1,2] ✉, Conall H. Watson[1], Mary E. Ramsay[1] & Nick J. Andrews[1,2]

An elevated risk of Guillain-Barré syndrome following respiratory syncytial virus vaccination in older adults was reported from the United States in the first year of their programme. This national study assesses the risk of Guillain-Barré syndrome (GBS) following vaccination against respiratory syncytial virus (RSV) using bivalent pre-F (Abrysvo, Pfizer) vaccine in England's programme for 2.5 million 75-79 year olds. The vaccination campaign began in September 2024, with initial risk assessed three months later using the intravenous immunoglobulin registry for the target age group. This ecological analysis used historical data and case-coverage comparisons linking to the national RSV vaccine register. A final six-month analysis combined the intravenous immunoglobulin and GBS-coded hospital discharge data, using both case-coverage and the planned self-controlled case-series method, focusing on the 0–42 day post-vaccination risk window. The initial analysis showed a significant increased risk of GBS after RSV vaccination, confirmed by both ecological and case-coverage methods. Final results were consistent, with the self-controlled case-series estimating a relative incidence of 3.34 (95% CI: 2.12–5.28) and an attributable risk of 23 (95% CI: 17–26) cases per million doses. Here, we show a small risk of GBS following RSV vaccination with the bivalent pre-F in England's older adults' programme, but this is at a level that is far exceeded by the vaccine benefits.

Guillain-Barré syndrome (GBS) is an Adverse Event of Special Interest in the RSV vaccine programme due to pre-licensure phase 3 clinical trials reporting a small number of cases of GBS in older adults[1,2] as well as historical evidence suggesting small elevated risks after some other respiratory vaccines such as influenza and COVID-19[3–5]. The USA introduced a vaccine programme a year before the UK with two vaccine products available, GSK's AS01$_E$ adjuvanted pre-F (Arexvy) and Pfizer's bivalent pre-F (Abrysvo).

GBS is an autoimmune disease that is usually preceded by a viral or bacterial infection. It is the commonest cause of acute neuromuscular paralysis in the UK with an estimated annual incidence of 1.5/100,000 (1.3–1.8)[6,7]. Cases occur across all ages but are more common

in older males. In the UK, neurologists diagnose it in secondary care using clinical, lab, and electrophysiological findings. For 25% of cases progressive weakness can lead to respiratory failure requiring ventilatory support[7]. Treatments for hospitalised cases may include plasma exchange and intravenous human immunoglobulin (IVIG).

Due to the significant burden of respiratory syncytial virus illness in the UK population[8] the UK's Joint Committee on Vaccines and Immunisation recommended RSV immunisation programmes to protect both infants and older adults, initially 75–79 years olds. These programmes were implemented on 1 September 2024, with a single dose for the older adults and an antenatal maternal dose for infant protection, both using bivalent pre-F vaccine (Abrysvo, Pfizer)[9]. From

---

[1]Immunisation and Vaccine Preventable Diseases Division, UK Health Security Agency, London, UK. [2]NIHR Health Protection Research Unit in Vaccines and Immunisation, London School of Hygiene & Tropical Medicine, London, UK. ✉e-mail: julia.stowe@ukhsa.gov.uk

implementation of the programme up until 31st March 2025 one and a half million doses were given to the older adult population.

During the monitoring in the first season the an FDA analysis of 3.2 million Medicare vaccine recipients did not identify a differential risk between the two vaccine products[10] and an independent analysis of 4.7 million vaccinees in the Epic Cosmos system reported 18.2 cases of GBS per million doses of Abrysvo (IRR: 2.4: 95%CI:1.5–4.0) and 5.2 cases per million doses of Arexvy (1.5; 95%CI:0.9–2.2)[11]. Methodological differences between the studies may account for the variance between the study results. Unlike the FDA study, the Epic study did not include chart confirmation of GBS cases, adjustment for seasonal variation in GBS incidence rates or deaths (eg, Farrington adjustments), and allowed for self-reported vaccine exposure data. In this study we initially used IVIG data indicated for GBS treatment from all NHS hospital trusts in England to carry out a rapid assessment of any RSV vaccine (Abrysvo, Pfizer) risk in the first 3 months after introduction (September 2024 to November 2025) using both an ecological and case-coverage approach. We then combined IVIG data with all hospitalisations with a GBS discharge code to assess the risk for the 6 month post introduction period (September 2024 to March 2025) using case-coverage and self-controlled case-series methods. This fuller case capture allowed for the estimation of an attributable risk based on the total vaccine doses given in England.

## Results

### Rapid assessment

Between 1st September 2024 and 30th November 2024 a total of 43 IVIG cases aged 74–79 were reported with the treatment date in this period. In the previous 14 quarters from March 2021 to August 2024 a total of 279 cases were reported (mean 20 per quarter). Fig. 1 shows the case numbers with the fitted trend line to the data prior to September 2024. The predicted number of cases was 23 meaning an excess of 20 cases in this period. This excess was statistically significant (incidence rate ratio 1.8 (95% confidence interval: 1.2–2.6, $p = 0.003$)). There was no evidence of seasonality in the data ($p = 0.26$).

For the case-coverage analysis of 43 cases 41 cases matched to IIS to obtain RSV vaccination status. Of these 19 cases were vaccinated with 17 within the previous 0–42 days. Each case was matched to vaccination coverage using the data in Fig. 1. The average matched population coverage of vaccination within the prior 6 weeks was 15.5%. The odds ratio for the risk of vaccination within 6 weeks in cases compared to the matched population was 4.27 (95% CI: 2.21–8.24). This means of the 17 cases an estimated $(3.27/4.27) \times 17 = 13$ were vaccine attributable.

Both of the rapid analyses, therefore, indicated a statistically significant increase in GBS risk following RSV vaccination.

### Final assessment

For the final assessment there were initially 139 IVIG cases with a treatment date from 1st September 2024 to 31st May 2025. 17 (12.2%) of these did not link to the IIS vaccine register. The remaining 122 were merged with an initial 265 SUS cases with onset from March 2024 to March 2025. 72 IVIG matched to a SUS episode and of those 50 IVIG cases without a SUS match 19 were matched to an ICD10 G00-G99 coded episode and 31 did not match to SUS. Of the 193 SUS episodes that did not match to IVIG 113 were dropped as they occurred prior to September 2024 leaving 80 cases only identified in SUS (Fig. 2). After creating the index date a further 29 IVIG cases were dropped for being outside the study period. This left a total of 173 cases of whom 80 were only in SUS, 65 were IVIG matched to a GBS SUS episode, 18 were IVIG matched to another neurological SUS episode, and 10 were IVIG with no SUS match. Only 3 cases had a Classification of Interventions and Procedures OPCS code for plasma exchange and all of these also had IVIG as a treatment.

Of the 173 cases a total of 83 were vaccinated by the index date with 48 of these being within 0–42 days of vaccination. The 173 cases are described in Table 1 by vaccination status and demographic factors. No demographic factors were found to be associated with being a case in the vaccine risk period, but there was a clear difference by diagnosis group with a larger proportion in the vaccine risk window for the IVIG cases with a SUS GBS code, a smaller proportion for the SUS only cases and a much lower proportion for the IVIG cases that had a different neurological code on discharge. Based on this observation a post-hoc analysis excluding GBS IVIG cases matching only to other SUS neurological conditions as well as also excluding those that did not match to a SUS admission episode. was also done. There were only 2 concomitant (±3 days) vaccinations with influenza vaccine and 1 with covid-19 vaccine, none of which were the same day. Overall, 88 (51%) had had a flu vaccine and 70 (40%) a covid-19 vaccine by the index date.

For the case-coverage analysis the 173 cases were matched to the coverage data in Fig. 3. Of the cases 48/173 (27.8%) were vaccinated within 42 days compared to the average matched coverage of 11.3%. This gave an odds ratio of 3.30 (95% CI: 2.32–4.69) and an attributable number of cases of 33.5 cases ($48 \times (2.3/3.3)$). For vaccination of more than 42 days prior, after excluding the risk period, the proportion vaccinated was 35/125 (28.0%) compared to an average matched coverage for vaccination over 42 days ago of 25.9%. This gives an odds

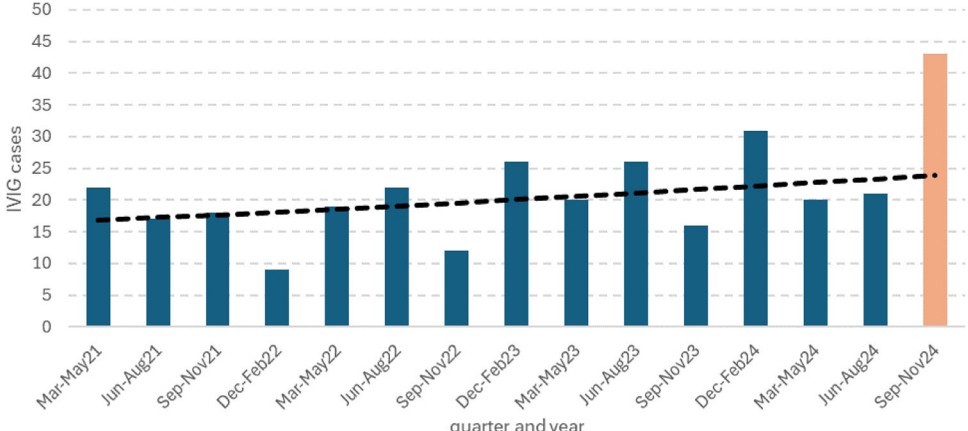

**Fig. 1 | Ecological assessment of IVIG case numbers in 74–79 year olds in England by quarter from March 2021 to November 2024.** Orange bar shows the post vaccine introduction period and the dashed line is the trend in case numbers prior to vaccination (prior to September 2024) fitted using Poisson regression to predict the expected cases for September-November 2024.

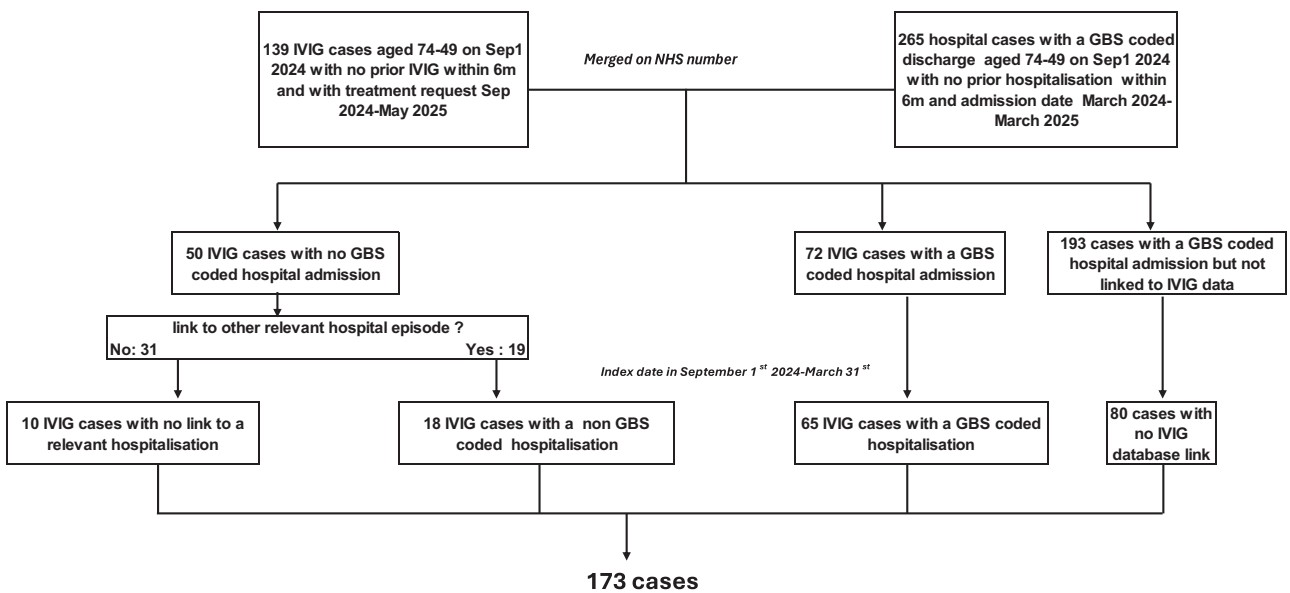

**Fig. 2** | Data flow chart of the 173 individuals in the final analysis.

ratio of 0.90 (95% CI: 0.58–1.40) and indicates no risk beyond 42 days. This provides validation for the use of this period as the control period in the SCCS analysis.

The self-controlled case-series analysis was based on the 83 vaccinated cases. Results are shown in Table 2 and the timing of cases in the 10 weeks after vaccination in Fig. 4. The main analysis gave a significantly raised relative incidence of 3.34 (95% CI: 2.12–5.28) with an estimated 33.6 attributable cases, very similar to the case-coverage odds ratio. Fig. 4 shows most of the risk occurring on days 8 to 21 post vaccination. There was no significant evidence of an interaction between vaccine risk and sex ($p = 0.09$) or risk group ($p = 0.08$). Removing the IVIG cases that only matched to a neurological ICD code increased the relative incidence to RI 3.78 (95% CI 2.34-6.11) as expected but left the attributable cases at a similar number. Additionally removing the IVIG cases that did not link to a SUS episode further increased the relative incidence RI (4.10 (95% CI 2.49-6.74)).

For the sensitivity analysis with the removal of cases with a previous episode within 12-months led to the removal of 2 cases in the background period and a small increase to the RI (3.58 (95% CI 2.25-5.69)).

The analysis assessing the risk of influenza and Covid-19 vaccines in the 65+ ages based on IVIG cases with a SUS GBS ICD discharge showed no risk for either vaccine. For influenza vaccine there were 70 cases with 15 in the risk window and a RI of 0.78 (95%CI: 0.44–1.38). For Covid-19 vaccine there were 60 cases with 13 in the risk window and a RI of 0.79 (95% CI: 0.43–1.46). As these vaccines show no evidence of a GBS risk adjustment for them in the analysis of the RSV vaccine risk was not done.

The total doses given by early March 2025 was 1,483,800. With 33.6 attributable cases (95% CI 25.4 to 38.9) this gives an attributable risk of 22.7 (95% CI: 17.1–26.2) per million doses (Table 2).

## Discussion

Both the initial rapid assessment and the final assessment after 6 months indicate a small but clear increased risk of GBS following RSV vaccination with bivalent pre-F in England's older adults' programme of around 23 cases per million doses. As well as the ecological spike in IVIG cases the analyses based on case-coverage and SCCS gave consistent results. The case-coverage and ecological analyses have the advantage of being able to be done rapidly, particularly when using IVIG cases. The SCCS needed a longer follow-up to allow person time to

accrue beyond the 6 week risk window and SUS hospital cases needed longer to allow for hospital discharge coding.

The case-coverage analysis also demonstrated that the increase of GBS vaccinated cases returned to baseline after 42 days when compared to the vaccine coverage estimates. Our data suggest case-patients typically present between days 8 and 21 post vaccination.

SUS GBS cases may include those treated with plasma exchange/plasmapheresis which is considered equivalent efficacy to IVIg therefore use for similar disease severity. However our searches for procedure codes indicating plasma exchange did not return any new information, all patients with a plasma exchange code were also recorded in the IVIg database. It is possible that GBS in SUS patients not appearing in the IVIg database may indicate less severe disease, in line with the clinical criteria for IVIG use but it is important to note that the IVIG and SUS datasets are independently maintained and linked via NHS number. To ensure accurate determination of vaccination status, all cases included in the analysis were required to link to the population denominator dataset and therefore possess a valid NHS number.

The biological mechanism for an association is unclear. In previous vaccine mediated GBS it has been suggested that immune stimulation and molecular mimicry could play a role in the pathogenesis or that it is a direct effect from a component of the vaccine[4,12]. When comparing cases in the risk period to cases that were outside the risk window or unvaccinated no risk factors were identified. The overall proportion of the GBS cases with any clinical risk factor was 66% (115/173) which was slightly higher than the general population aged 74–79 which was 60% as recorded in the IIS database.

Although in US the RSV vaccine is given to a wider age cohort (60 years and over), the attributable risk of 22.7 per million was similar to that reported in a US study for the same vaccine based on ICD coded hospital discharged GBS of 18.2 per million[11] but lower than the primary analysis from a US FDA study, restricted to adults 65 and older, in which case-confirmation was done where the AR was 9.0 per million doses[10]. It is interesting to note that the FDA study also performed an analysis on all cases without case confirmation and in this analysis the AR was 18.3 per million. The reason for the higher AR when using all cases was because the proportion of cases confirmed was lower if they occurred in the risk window (62%) compared to the control window (82%). This difference might have been chance as numbers were fairly small but if it is a real finding would point to the possibility of knowledge of recent vaccination status influencing the coding of cases as

**Table 1 | Description of 173 individuals in the final analysis by vaccination status using Fisher's exact test comparing proportions in the risk period to the combined vaccinated and non-risk interval**

| Factor | Level | unvaccinated | Vaccination status | | Total | % within 0–42 days | p value* |
| | | | >42 days prior | within 0–42 days | | | |
|---|---|---|---|---|---|---|---|
| Total | all | 90 | 35 | 48 | 173 | 28% | |
| Age Sep 1 2024 | 74 | 17 | 2 | 5 | 24 | 21% | 0.52 |
| | 75 | 13 | 4 | 9 | 26 | 35% | |
| | 76 | 18 | 7 | 8 | 33 | 24% | |
| | 77 | 15 | 10 | 15 | 40 | 38% | |
| | 78 | 8 | 6 | 5 | 19 | 26% | |
| | 79 | 19 | 6 | 6 | 31 | 19% | |
| Sex | Female | 36 | 11 | 23 | 70 | 33% | 0.23 |
| | Male | 54 | 24 | 25 | 103 | 24% | |
| Immunosuppressed | no | 80 | 30 | 44 | 154 | 29% | 0.60 |
| | yes | 10 | 5 | 4 | 19 | 21% | |
| Respiratory condition | no | 74 | 30 | 42 | 146 | 29% | 0.64 |
| | yes | 16 | 5 | 6 | 27 | 22% | |
| Kidney disease | no | 71 | 29 | 41 | 141 | 29% | 0.51 |
| | yes | 19 | 6 | 7 | 32 | 22% | |
| Heart condition | no | 56 | 16 | 30 | 102 | 29% | 0.61 |
| | yes | 34 | 19 | 18 | 71 | 25% | |
| Asplenia | no | 90 | 35 | 47 | 172 | 27% | 0.28 |
| | yes | 0 | 0 | 1 | 1 | 100% | |
| Liver | no | 88 | 35 | 48 | 171 | 28% | 1.00 |
| | yes | 2 | 0 | 0 | 2 | 0% | |
| Endocrine | no | 67 | 24 | 38 | 129 | 29% | 0.44 |
| | yes | 23 | 11 | 10 | 44 | 23% | |
| Any | no | 32 | 7 | 19 | 58 | 33% | 0.37 |
| | yes | 58 | 28 | 29 | 115 | 25% | |
| Died within 60 days | no | 84 | 30 | 41 | 155 | 26% | 0.28 |
| | yes | 6 | 5 | 7 | 18 | 39% | |
| Diagnosis group | Immunoglobulin and hospitalisation with GBS code | 28 | 6 | 31 | 65 | 48% | <0.001 |
| | Immunoglobulin and hospitalisation with other neurological code | 12 | 5 | 1 | 18 | 6% | |
| | Immunoglobulin but no current link to a hospitalisation | 4 | 3 | 3 | 10 | 30% | |
| | Hospitalisation with GBS code but no link to immunoglobulin | 46 | 21 | 13 | 80 | 16% | |

*Fisher's exact test comparing proportions in the risk period.

GBS when in fact they are not, or potentially if post-vaccination GBS presented less severely or atypically, leading to fewer being confirmed at case note review.

In our study we did not rely solely on GBS coded cases as we also used IVIG treatment but if the positive predictive value of GBS cases in the risk and control windows had been the same as in the FDA study then the AR would have reduced to 12.1 per million (Supplementary material). A limitation of estimating the AR based on validated cases only is that many cases cannot be fully validated and are excluded because of limited information (i.e. Brighton level 4 certainty[13]) rather than because they are clearly not GBS (Brighton level 5) leading to underestimation of AR through lack of sensitivity. On the other-hand true differential positive predictive values of background cases in risk window compared to the control window can lead to over estimation of risk, as can attributing a non-GBS vaccine induced adverse event as GBS. Non differential PPV with respect to background cases will not give any bias. Whilst we did not validate cases in our study it is possible differential positive predictive values with respect to background

cases were present due to awareness of the hypothesis as GBS was mentioned as an important potential risk following two GBS/Miller-Fisher variant cases in patient information and the Public Assessment Report[14]. There were also reasons to suggest this possible bias is not large in our study. First, the IVIG GBS cases that were discharged with a non-GBS code did not show any risk in the SCCS analysis, second the GBS cases clustered specifically in the second and third weeks after vaccination which is most biologically plausible for an immune-mediated reaction and third no risk was seen after influenza or COVID-19 vaccines for which a similar (if perhaps smaller) bias may have been expected.

Recent studies have shown clear impact and high effectiveness of the bivalent pre-F vaccine in the older adults in the UK[8,15,16]. The risk of about 23 GBS per million doses of bivalent pre-F is small compared to vaccine benefits which were conservatively estimated at 1128 prevented hospital admissions and 86 prevented deaths per million doses in analyses presented in June 2025 to the UK Joint Committee on Vaccine and Immunisation[17]. Early medical intervention for GBS may

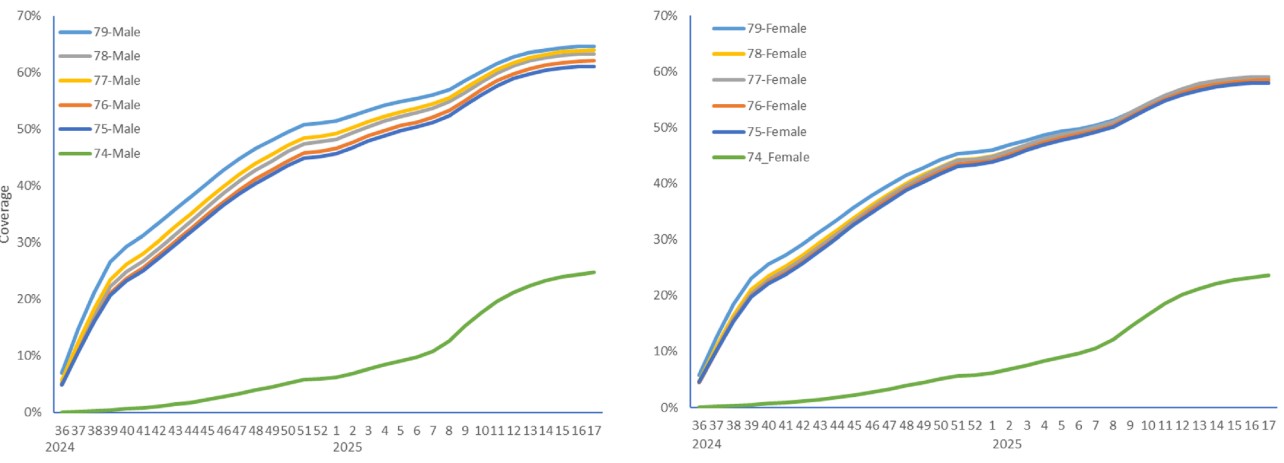

**Fig. 3 | RSV vaccine coverage in 74-79 year olds in England.** Using data from the Immunisation Information System (IIS) by age at September 1st 2024, sex and week number.

**Table 2 | Self-controlled case-series results: Relative incidence (95% confidence interval) of GBS in the period after vaccination and attributable risk by analysis type**

| Analysis | Period after vaccination (days) | Cases | Mean person-time (days) | Relative incidence (95% CI) | Attributable cases[a] | Attributable risk per million doses (95% CI)[b] |
|---|---|---|---|---|---|---|
| Main | 0–42 | 48 | 41.8 | 3.34 (2.12–5.28) | 33.6 | 22.7 (17.1–26.2) |
| | 43+ | 35 | 117.7 | | | |
| Removing IVIG-other neurological | 0–42 | 47 | 41.7 | 3.78 (2.34–6.11) | 34.6 | 23.3 (18.1–26.5) |
| | 43+ | 30 | 115.7 | | | |
| Removing IVIG-other neurological and IVIG not matched to SUS | 0–42 | 44 | 41.8 | 4.10 (2.49–6.74) | 33.3 | 22.4 (17.7–25.3) |
| | 43+ | 27 | 118.4 | | | |
| Removing if prior episode within 12 months | 0–42 | 48 | 41.8 | 3.58 (2.25–5.69) | 34.6 | 23.3 (18.0–26.9) |
| | 43+ | 33 | 118.2 | | | |

[a]Cases x (RI-1)/RI.
[b]Based on 1,483,800 vaccine doses given in England by early March 2025.

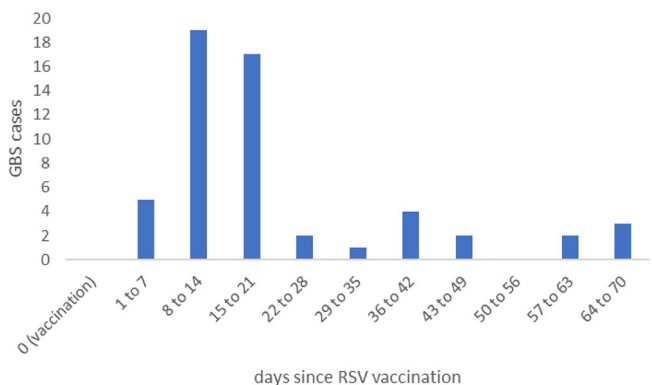

**Fig. 4 |** Distribution of GBS cases by days from RSV vaccination in the 10 weeks after vaccination.

reduce severity and improve outcomes: the UK's Medicines and Healthcare products Regulatory Agency has advised health professionals inform vaccinees to seek medical assessment for any early signs and symptoms[18].

RSV vaccination in the UK has now been recommended for those aged over 80[18]. If the risk of post-vaccination GBS is proportional to the background incidence, which is plausible[19] then the vaccine attributable risk may be higher in this age group, but the benefits are also likely to be higher in this group due to the higher incidence of RSV

hospitalisations and deaths[8]. This will require further study with the roll-out to this age group.

In conclusion our study demonstrates a small risk of GBS following RSV vaccination with the bivalent pre-F in England's older adults' programme, but this is at a level that is far exceeded by the vaccine benefits.

## Methods
### Data sources
The data sources used for the analysis covered the whole of England and comprised of the National Immunoglobulin Database (IVIG), the Secondary User Service (SUS) database and the Immunisation Information System (IIS). It was not possible to perform a case note review on these cases. UKHSA has legal permission, provided by Regulation 3 of The Health Service (Control of Patient Information) Regulations 2002, to process patient confidential information for national surveillance of communicable diseases and as such, individual patient consent is not required. Separate ethics approval is not required for the work.

### National immunoglobulin (IVIG) database
The National Immunoglobulin Database[20] managed by Medical Data Solution and Services (MDSAS) provided individual level data on the use of immunoglobulin as requested by hospital trusts in England. Patients are eligible for NHS IVIg if they have a diagnosis of GBS and at least one of severe disease (Hughes grade 4 or 5, bedbound/chairbound or ventilated), signs of progression towards respiratory failure

and ventilation (clinical assessment or mEGRIS ≥ 3), or poor prognosis (mEGOS ≥ 4)[21]. Cases where the indication for treatment was GBS were sent monthly to UKHSA from MDSAS and covered all ages with data going back to 2020. When an individual had an intravenous immunoglobulin (IVIG) treatment within 6 months of a previous treatment this was removed as it was assumed to be part of the same episode.

### SUS database

The SUS database comprises of ICD10 coded admitted patient care episodes from NHS hospitals in England. In order to exclude historical recording of a GBS diagnosis in the hospital record, all discharges where there was an ICD code for GBS (G610) in one of the first 3 discharge codes positions for the period March 1st 2024 to March 31st 2025 were retained as long as there was no prior episodes within 6 months. Only cases aged 74–79 on September 1st 2024 were included. Note that those aged 74 were included because they become eligible as they turn 75. These data were extracted in June 2025 to allow good completeness of hospital admissions to the end of March 2025 since the coding only occurs on discharge meaning there is a lag in the data. Due to this data lag some cases in the IVIG may therefore not appear in the SUS data, particularly if they have a long hospital stay.

### Immunisation Information System (IIS) database

The IIS contains individual RSV, influenza and COVID-19 vaccine histories and includes batch and manufacturer information[22] alongside a whole England population denominator that contains information on date of birth, death date, sex and COVID-19 vaccination eligibility risk group status according to the Green Book definition[23].

### Data linkage

For the initial rapid analysis, IVIG cases to the end of November 2024 aged 74–79 on September 1st 2024 were linked to the IIS using NHS number. Cases that did not link were excluded from the case-coverage analysis (linkage rate 91.3%) on IVIG cases but retained for the ecological analysis. The index date used in this analysis was date of IVIG treatment request.

For the final analysis using IVIG and SUS cases those IVIG cases aged 74–79 that linked to IIS were retained as well as all SUS cases aged 74-79 that linked to IIS (linkage rate 99.1%). The IVIG cases were merged with the SUS cases and where cases occurred in both databases the earliest of the date of hospitalisation or IVIG treatment was used as the index date unless the gap was greater than +/-6 months between hospitalisation and IVIG, in which case they were regarded as separate episodes. IVIG cases that did not match to a SUS GBS coded hospitalisation were matched to SUS to identify the reason for that admission from the ICD10 coding. If there was a ICD discharge code in Chapter VI G00-G99 Diseases of the nervous system within 35 days this admission date was used as the index date. For IVIG cases not matching to a hospitalisation the index date used was 7 days prior to the IVIG treatment request date based on this being the median interval for those where a hospital admission date was known. After this linkage the cases were restricted to those with an index date from September 1st 2024 to March 31st 2025 (Fig. 2).

### Statistical methods

#### Descriptive analysis.
IVIG Cases by quarter (Sep-Nov, Dec-Feb, Mar-May, Jun-Aug) from March 2021 to November 2024 were plotted and a time trend using the data prior to September 2024 fitted using Poisson regression to predict the expected cases for September-November 2024. The observed cases are compared ecologically to assess any excess. The significance of the excess was obtained from the model by

using an indicator variable for the vaccine period. This model was also used to check for any seasonality by quarter.

The IVIG/SUS cases for the analysis from 1st September 2024 to 31st March 2025 are described by vaccination status (unvaccinated, vaccinated within 42 days and vaccinated more than 42 days prior) and age, sex, risk group, death within 60 days, and type of case (IVIG and SUS (ICD10 G610: GBS), IVIG and SUS (ICD10 G00-G99: Chapter VI Diseases of the Nervous System), IVIG (no SUS link) and SUS only). The percentage in the risk window are compared by these factors using a Fisher's exact test. Numbers of co-administrations of influenza and COVID-19 vaccines within ±3 days and receipt of these vaccines at any time prior to the index date are also given.

#### Case-coverage analysis.
The case-coverage analysis[24] used the national RSV vaccine data and denominator data from the IIS to derive vaccine coverage for the RSV vaccination programme. The coverage numerator was stratified by age on 1st September 2024 (74–79 years), sex and week of RSV vaccine administration. The coverage denominator was stratified by age on 1st September 2024 and sex.

For case-coverage the risk period of interest was vaccination within 6 weeks prior to the index date although the period where vaccination was over 6 weeks before the index date was also checked to ensure the proportion vaccinated in this period was close to that expected assuming no risk.

For the rapid analysis using the IVIG cases to November 2024 each IVIG case was matched to this IIS coverage based on week of treatment as the index date, age as on 1st September 2024 and sex. The cases were also matched to coverage from 6 weeks before IVIG treatment to enable estimation of coverage in past 6 weeks. For the final analysis using SUS and IVIG cases the same is done but matching was based on the week of the index date from the combined data.

The odds of vaccination in the previous 6 weeks in cases compared to the matched coverage was calculated using conditional logistic regression with vaccination status of the cases as the outcome and an offset for the logit of the matched coverage. To check for any risk beyond 6 weeks the same analysis was done assessing the proportion vaccinated more than 6 week prior to the index date compared to the equivalent coverage, after removal of those vaccinated in the 6 week risk period.

#### Self-Controlled Case-Series (SCCS).
The Self-Controlled Case-Series method[25] was used to test the hypothesis of an increased risk of GBS in the period 0–42 days after vaccination. This is the primary analysis. Only vaccinated cases were included and only person time after vaccination used, meaning the control period was all person time from 43 days after vaccination to the end of the study period (March 31 2025). This vaccine only analysis was done because a recent GBS diagnosis may delay RSV vaccination or mean it is less likely to be given at all.

Due a high level of collinearity between vaccine period and calendar period no seasonal adjustment was done on the basis that no seasonality was apparent when assessing the IVIG cases by quarter in the descriptive analysis. Evidence for any interaction between sex and vaccine risk as well as being in a clinical risk group and vaccine risk was assessed.

To check for the possibility of confounding effects for any risk due to COVID-19 or Influenza vaccines analyses are also done within just those SUS cases who also had IVIG at the time of the episode for those aged 65 and over in the period September 1st 2024 to March 31st 2025. This also used a 42 days risk window with the index date being hospital admission date and just within those vaccinated, so using person time beyond 42 days as the comparator.

A sensitivity analysis was also performed where any cases with a previous episode within 12 months, rather than just within 6 months, were removed.

**Attributable risk estimation.** Attributable risk was calculated based dividing the estimated number of cases attributable by the total doses given by early March 2025 (week 10 2025). This date was used to ensure the majority of vaccinees had time for GBS to occur in the main risk period of a few weeks after vaccination. The number of attributable cases was calculated as observed cases in the risk window x (odds ratio-1)/odds ratio for case-coverage and observed cases in the risk window x (relative incidence -1)/relative incidence for the SCCS study.

### Reporting summary

Further information on research design is available in the Nature Portfolio Reporting Summary linked to this article.

## Data availability

This work is carried out under Regulation 3 of The Health Service (Control of Patient Information; Secretary of State for Health, 2002) using patient identification information without individual patient consent as part of the UKHSA legal requirement for public health surveillance and monitoring of vaccines. As such, authors cannot make the underlying dataset publicly available for ethical and legal reasons. However, all the data used for this analysis is included as aggregated data in the manuscript tables and appendix. Applications for relevant anonymised data should be submitted to the UKHSA Office for Data Release at https://www.gov.uk/government/publications/accessing-ukhsa-protected-data.

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

## Acknowledgments

The authors reported there is no funding associated with the work featured in this article.

## Author contributions

J.S.: Conceptualisation, Data curation, Methodology, Writing–original draft, Writing—review and editing. N.J.A.: Conceptualisation, Formal analysis, Methodology, Writing—original draft. C.H.W.: Conceptualisation, Methodology, Writing—original draft. M.E.R.: Conceptualization, Methodology, Writing—review and editing.

## Competing interests

The Immunisation and Vaccine Preventable Diseases Department at UKHSA has provided vaccine manufactures with post-marketing

surveillance reports, which the Marketing Authorization Holders are required to submit to the UK Licensing authority in compliance with their Risk Management Strategy. A cost recovery charge is made for these reports. The authors declare no other competing interests.
