## [Transparent Peer Review file · Nature Communications]

Assessing the risk of Guillain-Barré syndrome in older adults after bivalent RSV pre-F vaccination in England

Corresponding Author: Dr Julia Stowe

Version 0:

Reviewer comments:

Reviewer #1

(Remarks to the Author)

I commend the authors on an important study that will be a useful contribution to the literature. I have some specific largely methodological comments to improve the usefulness of the manuscript.

1. In the introduction, the authors cite the study by Fry et al and suggest that they identified a differential risk of GBS by vaccine. No head-to-head statistical comparison was completed as part of the study between the brands, and, the IRRs have overlapping confidence intervals. Suggest removing the implication that a difference was found and just state the results, including IRRs which are less impacted by GBS background rates in specific vaccinee populations (see discussion in Lloyd et al), and include mention of methodological differences that may account for the variance in the results (such as no chart-confirmed GBS; no adjustment for deaths, vaccine brand misclassification).

a. Current text: While the FDA analysis of 3.2 million Medicare vaccine recipients did not identify a differential risk between the two vaccine products (11), an independent analysis of 4.7 million vaccinees in the Epic Cosmos system reported 18.2 cases of GBS per million doses of Abrysvo and 5.2 cases per million doses of Arexvy (12).

b. Suggest alternate text: An FDA analysis of 3.2 million Medicare vaccine recipients did not identify a differential risk between the two vaccine products (11), and an independent analysis of 4.7 million vaccinees in the Epic Cosmos system reported 18.2 cases of GBS per million doses of Abrysvo (IRR: 2.4; 95%CI:1.5–4.0) and 5.2 cases per million doses of Arexvy (1.5; 95%CI:0.9–2.2) (12). Methodological differences between the studies may account for the variance between the study results. Unlike the FDA study, the Epic study did not include chart confirmation of GBS cases, adjustment for seasonal variation in GBS incidence rates or deaths (eg, Farrington adjustments), and allowed for self-reported vaccine exposure data.

2. For this study, the clean window to define incident GBS cases is six months, not one year (page 4), which is different from other studies, which generally use a 1-year clean window. A sensitivity analysis with a one-year clear window should be performed if feasible. This should be mentioned as a limitation if no sensitivity analysis is feasible.

3. The GBS cases were defined in different ways: IVIG only, IVIG+GBS dx, IVIG+any neurologic disease and GBS dx without IVIG. Different case definitions were used in different analyses. Not all IVIG cases may represent GBS, as patients with other conditions may also receive IVIG. However, IVIG cases without a corresponding SUS GBS dx were also included in the analysis.

a. As IVIG can be used for non-GBS conditions, a sensitivity analysis removing IVIG only cases should be added to Table 2.

4. In Table 2, the number of excess cases should be presented per 1 million doses to facilitate comparison with findings from other studies or vaccines. This approach also helps prevent readers from confusing the number of attributed cases across the whole season with the attributable risk per 1 million doses. Currently, this data is in the text (22.7 (95% CI: 17.1-26.2) per million doses) but not included in Table.

a. Ideally, rates for both cases per X person years and X doses would be included as was done in Lloyd et al.

5. The control GBS rate was approximately 7.3 cases per 100,000 person-years (35/ [1,483,800*117.7/365]), closely aligning with data reported by the FDA for ABRYVO recipients. However, the incidence IRRs and AR observed here were notably higher. This discrepancy may be in part due to some methodological differences such as the absence of chart confirmation for GBS cases, application of a shorter clear window (previously mentioned), and absence of seasonality and Farrington adjustments. Farrington adjustment accounts for the potential violation of the SCCS assumption with the event-dependent

observation period. Additional sensitivity analyses should be included in Table 2 on this basis. The authors applied FDA's PPVs in both risk and control periods, and the derived IRR and AR that were broadly comparable to those in the FDA's study, but this data is only included in the supplement.

- a. Re absence of chart confirmation for GBS cases, suggest including the PPV sensitivity results to Table 2
- b. Suggest including Farrington adjustment. If Farrington adjustment is not feasible, please add a sensitivity analysis removing patients who died within 30 days of GBS to evaluate the potential violation of assumption with the event-dependent observation period.
- c. Sensitivity analysis removing the GBS cases defined by IVIG but without a link to a hospitalization should be added (mentioned above)
- d. Sensitivity analysis using one-year clear window to exclude potential prevalent GBS cases (mentioned above) (versus 6 months in primary analysis)
- e. Seasonality adjustment – it is mentioned that no seasonal trend was seen in quarterly GBS cases. Wondering if this would be seen if you examined monthly trends. If these are presented, authors should add seasonality adjustment to Table 2 as well.

6. Clarify how the incidence rate ratio was calculated—was it derived using a Poisson regression model?

Example: This excess was statistically significant (incidence rate ratio 1.8; 95% CI: 1.2–2.6; $p = 0.003$).

7. Re this sentence: “Both of the rapid analyses therefore indicated a significant risk of GBS following RSV vaccination.”

Suggest clarifying significant means statistically significant as follows: “Both of the rapid analyses, therefore, indicated a statistically significant increase in GBS risk following RSV vaccination.”

8. It is known that GBS risk increases with age (Sejvar JJ, Baughman AL, Wise M, Morgan OW. Population incidence of Guillain-Barré syndrome: a systematic review and meta-analysis. *Neuroepidemiology* 2011; 36(2): 123-33). No age-stratified estimates of GBS risk after RSV vaccination are currently available from the US, where RSV vaccination has been in use among a broader age cohort (persons 60 years and older) to allow comparison of the measured risk in the same age group as vaccinated in the UK. Suggest adding some discussion of this point.

(Remarks on code availability)

Reviewer #2

(Remarks to the Author)

This is a well-done analysis that contributes meaningfully to the RSV vaccine literature on a topic of great interest to the international vaccine community. To my knowledge these data represent the first data from any country outside of the US powered to analyze the potential relationship between older adult RSV vaccination and vaccine-associated GBS. The findings in this manuscript replicate what has been demonstrated in the US and therefore represent a major finding in post-licensure vaccine safety surveillance. These data help support what was observed both in the phase 3 trials and now in post-licensure studies in the US—that there is likely a relationship between RSV vaccination and GBS.

Please note the following comments:

Intro:

32: consider moving line 39 to start, then intro about GBS and GBS as an adverse event of special interest.

48: Risk of GBS after COVID-19 has not been consistent. Not sure would say resp virus vaccines generally. Could consider mentioning shingrix.

57: might note this analysis did NOT have chart confirmation. This is an important reason the absolute magnitude of AR is likely larger.

59: would highlight here when referencing RSV vaccine risk you only have Pfizer (just to make clear this is not going to look at differential risk).

Methods:

112: what do you know about those that received IVIG without hospitalization? That seems somewhat strange.

133/34: are you able to comment on denominator data? Is there any bias in whether more likely to accurately report receipt or non-receipt of RSV vaccine? E.g. is no record of RSV vaccine in the IIS a good indicator RSV vaccine was not received? Is it possible less likely to have reported RSV vax status in unvaccinated with GBS?

159/160: one of the other central benefits of the SCCS is that it controls for intra-person confounders; assuming that those choosing to get vaccinated might be meaningfully different than those that do not.

Results:

182: The COVID-19 pandemic makes this ecologic analysis somewhat challenging. We know that circulation of other important viruses decreased during COVID and other viruses are important causes of GBS. Do you have the possibility of looking at any historical data from before the pandemic? If these data are not available please consider mentioning this in the discussion.

211-215: can you discuss potential reasons for this in the discussion? It would seem that SUS without IVIG would be less severe GBS. Is that the case? Or is it the case that IVIG was missed or not recorded? Is this suggesting that the vaccine-associated GBS was more likely to receive IVIG/be more severe? Alternatively, would severe GBS be more likely to have antecedent vaccine noted?

Discussion:

Important to note that the magnitude of the attributable risk per million doses is based on the number of observed/counted cases. In a situation in which this number is inflated (and based on US PPV analyses it is unlikely to be 100%) the absolute magnitude of the AR will always be higher in this type of analysis without chart confirmation. The IRR may be accurate but the absolute value, which is very important in risk-benefit discussions, is likely somewhat of an overestimate. Please consider discussing this further. The supplemental analysis is a nice addition, but of course the PPV is dependent on many things (health systems, knowledge of GBS/RSV association in the local setting, etc).

Might also note any findings related to maternal program to date in discussion of age-dependent risk (pregnant risk likely to be lower).

246: small is relative. Small in absolute terms compared to risk/benefit, but perhaps large in vaccine associated risk terms (eg compared to shingrix, pandemic flu, seasonal flu). Therefore may wish to say a small risk compared to benefit?

259: consider a citation

261: would be interested to learn more about risk factors considered in risk and control AND unvaccinated patients--were vaccinated and unvaccinated similar? This would be a nice addition if there is space.

(Remarks on code availability)

Version 1:

Reviewer comments:

Reviewer #1

(Remarks to the Author)

I have reviewed the updated version of the manuscript and the response to reviewer comments and do not have additional comments.

(Remarks on code availability)

Reviewer #2

(Remarks to the Author)

Authors have adequately addressed reviewer comments. This analysis will contribute meaningfully to our understanding of RSV vaccine safety and has important policy implications.

Follow-up comments:

In the discussion authors have added "Although in US the RSV vaccine is given to a wider age cohort (60 years and over)..." and then discuss the EPIC and FDA studies. Please be aware the FDA study was restricted to adults 65 and older (on Medicaid).

You note that AR is only impacted by inflated case counts if PPV is differential. I am not sure I understand this because the difference is absolute. Eg 4 cases in risk/2 cases in control leads to an AR of 2 whereas 8 cases in risk/4 cases in control leads to an AR of 4. These both have an IRR of 2 but the absolute attributable risk is different which means if there is overestimation of both cases and controls (even if overestimation is non-differential) the absolute risk may be different.

(Remarks on code availability)

REVIEWER COMMENTS

Reviewer #1 (Remarks to the Author):

I commend the authors on an important study that will be a useful contribution to the literature. I have some specific largely methodological comments to improve the usefulness of the manuscript.

1. In the introduction, the authors cite the study by Fry et al and suggest that they identified a differential risk of GBS by vaccine. No head-to-head statistical comparison was completed as part of the study between the brands, and, the IRRs have overlapping confidence intervals.

Suggest removing the implication that a difference was found and just state the results, including IRRs which are less impacted by GBS background rates in specific vaccinee populations (see discussion in Lloyd et al), and include mention of methodological differences that may account for the variance in the results (such as no chart-confirmed GBS; no adjustment for deaths, vaccine brand misclassification).

- a. Current text: While the FDA analysis of 3.2 million Medicare vaccine recipients did not identify a differential risk between the two vaccine products (11), an independent analysis of 4.7 million vaccinees in the Epic Cosmos system reported 18.2 cases of GBS per million doses of Abrysvo and 5.2 cases per million doses of Arexvy (12).
- b. Suggest alternate text: An FDA analysis of 3.2 million Medicare vaccine recipients did not identify a differential risk between the two vaccine products (11), and an independent analysis of 4.7 million vaccinees in the Epic Cosmos system reported 18.2 cases of GBS per million doses of Abrysvo (IRR: 2.4; 95%CI:1.5–4.0) and 5.2 cases per million doses of Arexvy (1.5; 95%CI:0.9–2.2) (12). Methodological differences between the studies may account for the variance between the study results. Unlike the FDA study, the Epic study did not include chart confirmation of GBS cases, adjustment for seasonal variation in GBS incidence rates or deaths (eg, Farrington adjustments), and allowed for self-reported vaccine exposure data.

-amended thank you

2. For this study, the clean window to define incident GBS cases is six months, not one year (page 4), which is different from other studies, which generally use a 1-year clean window. A sensitivity analysis with a one-year clear window should be performed if feasible. This should be mentioned as a limitation if no sensitivity analysis is feasible.

We have identified and investigated the cases that would have been removed if a 1-year clean window would have been applied and from the 173 cases in the full dataset 10 would have been removed.

We carried out a sensitively analysis and of these 10 two were vaccinated both 57 days before their GBS date and that is outside the 0-42 day risk window. After removing these cases the overall RI removing those 2 increases slightly to 3.58 (2.25-5.69).

This analysis has been added to the methods and results.

3. The GBS cases were defined in different ways: IVIG only, IVIG+GBS dx, IVIG+any neurologic disease and GBS dx without IVIG. Different case definitions were used in different analyses. Not all IVIG cases may represent GBS, as patients with other conditions may also receive IVIG. However, IVIG cases without a corresponding SUS GBS dx were also included in the analysis.

a. As IVIG can be used for non-GBS conditions, a sensitivity analysis removing IVIG only cases should be added to Table 2.

Just to confirm an analysis removing IVIG-other neurological conditions was performed and is reported (table 2) and commented on – “Removing the IVIG cases that only matched to a neurological ICD code increased the relative incidence as expected but left the attributable cases at a similar number”.

The remaining 10 cases that did not link to SUS will all have been recorded as having IVIG for treatment of GBS as this was the criteria for inclusion in our analysis (see methods).

Of these 10, 6 were vaccinated of which 3 were in the risk period (see table 1). We have now carried out a sensitivity analysis and if these 10 cases were removed the RI changes to 3.57, 95% (CI 2.23- 5.73).

We added this to the results and to table 2.

4. In Table 2, the number of excess cases should be presented per 1 million doses to facilitate comparison with findings from other studies or vaccines. This approach also helps prevent readers from confusing the number of attributed cases across the whole season with the attributable risk per 1 million doses. Currently, this data is in the text (22.7 (95% CI: 17.1-26.2) per million doses) but not included in Table.

a. Ideally, rates for both cases per X person years and X doses would be included as was done in Llyod et al.

This has been added to table 2 for per million doses. We don't think rates per million person years is a very useful metric. The information in the table could allow someone to calculate this as mean person time in the interval is given.

5. The control GBS rate was approximately 7.3 cases per 100,000 person-years (35/[1,483,800*117.7/365]), closely aligning with data reported by the FDA for ABRYSSVO recipients. However, the incidence IRRs and AR observed here were notably higher. This discrepancy may be in part due to some methodological differences such as the absence of chart confirmation for GBS cases, application of a shorter clear window (previously mentioned), and absence of seasonality and Farrington adjustments. Farrington adjustment accounts for the potential violation of the SCCS assumption with the event-dependent observation period. Additional sensitivity analyses should be included in Table 2 on this basis. The authors applied FDA's PPVs in both risk and control periods, and the derived IRR and AR that were broadly comparable to those in the FDA's study, but this data is only included in the supplement.

a. Re absence of chart confirmation for GBS cases, suggest including the PPV sensitivity results to Table 2

We would prefer to keep this analysis in the discussion. Unlike the other analyses added to table 2 this is not a sensitivity analysis like the others so would not have any actual data to show but just the calculated RI and AR. This analysis is also specifically done in the context of the US paper so seems more appropriate as a discussion point.

b. Suggest including Farrington adjustment. If Farrington adjustment is not feasible, please add a sensitivity analysis removing patients who died within 30 days of GBS to evaluate the potential violation of assumption with the event-dependent observation period.

Actually, because we only have one vaccine dose and only use post vaccine person time it is not necessary to use the event dependent method or worry about deaths. We checked this is correct with the first author of the paper on the SCCS that we reference (Heather Whitaker).

c. Sensitivity analysis removing the GBS cases defined by IVIG but without a link to a hospitalization should be added (mentioned above)

This has been added

d. Sensitivity analysis using one-year clear window to exclude potential prevalent GBS cases (mentioned above) (versus 6 months in primary analysis)

This has been added

e. Seasonality adjustment – it is mentioned that no seasonal trend was seen in quarterly GBS cases. Wondering if this would be seen if you examined monthly trends. If these are presented, authors should add seasonality adjustment to Table 2 as well.

Number are small by month which is why looked at quarter to see if there was seasonality. Within the monthly data there was no clear seasonal trend by month in the GBS cases.

6. Clarify how the incidence rate ratio was calculated—was it derived using a Poisson regression model?

Example: This excess was statistically significant (incidence rate ratio 1.8; 95% CI: 1.2–2.6; $p = 0.003$).

Yes it was Poisson as described in statistical methods under descriptive analysis (paragraph 1).

7. Re this sentence: “Both of the rapid analyses therefore indicated a significant risk of GBS following RSV vaccination.” Suggest clarifying significant means statistically significant as follows: “Both of the rapid analyses, therefore, indicated a statistically significant increase in GBS risk following RSV vaccination.”

Amended

8. It is known that GBS risk increases with age (Sejvar JJ, Baughman AL, Wise M, Morgan OW. Population incidence of Guillain-Barré syndrome: a systematic review and meta-analysis. *Neuroepidemiology* 2011; 36(2): 123-33). No age-stratified estimates of GBS risk after RSV vaccination are currently available from the US, where RSV vaccination has been in use among a broader age cohort (persons 60 years and older)

to allow comparison of the measured risk in the same age group as vaccinated in the UK. Suggest adding some discussion of this point.

The wider age group is now acknowledged in the discussion when comparing to the US studies.

Reviewer #2 (Remarks to the Author):

This is a well-done analysis that contributes meaningfully to the RSV vaccine literature on a topic of great interest to the international vaccine community. To my knowledge these data represent the first data from any country outside of the US powered to analyze the potential relationship between older adult RSV vaccination and vaccine-associated GBS. The findings in this manuscript replicate what has been demonstrated in the US and therefore represent a major finding in post-licensure vaccine safety surveillance. These data help support what was observed both in the phase 3 trials and now in post-licensure studies in the US--that there is likely a relationship between RSV vaccination and GBS.

Please note the following comments:

Intro:

32: consider moving line 39 to start, then intro about GBS and GBS as an adverse event of special interest.

Paragraph 3 on GBS being an AESI has been moved to the start as suggested

48: Risk of GBS after COVID-19 has not been consistent. Not sure would say resp virus vaccines generally. Could consider mentioning shingrix.

We do say "some other respiratory vaccines". The UK data has reported a risk after the AZ COVID vaccine and a seen after flu vaccines (seasonal and pandemic H1N1) has also been reported so we feel framing the question in terms of respiratory virus if fair.

57: might note this analysis did NOT have chart confirmation. This is an important reason the absolute magnitude of AR is likely larger.

This is discussed in relation to other studies in the discussion in paragraph 4 but the following has now been added to the Data Sources section in the methods "It was not possible to perform a case note review on these cases". We do also discuss what affect would occur if chart review gave a similar PPV to that found in the FDA US study.

59: would highlight here when referencing RSV vaccine risk you only have Pfizer (just to make clear this is not going to look at differential risk).

This has been added

Methods:

112: what do you know about those that received IVIG without hospitalization? That seems somewhat strange.

From the 173 cases in the final analysis there were 10 IVIG with no hospital (SUS) record. We did try to investigate with the data provider, but this was not possible. It is

possible some of these cases that had IVIG towards the end of the study period were not yet in the hospital discharge data which has a greater data lag. In response to review 1 we do now also do a sensitivity analysis excluding these cases.

133/34: are you able to comment on denominator data? Is there any bias in whether more likely to accurately report receipt or non-receipt of RSV vaccine? E.g. is no record of RSV vaccine in the IIS a good indicator RSV vaccine was not received?

In the Immunisation Information System (IIS) the denominator data is an independent data source from the IIS RSV vaccine data. It includes everyone in England that has an NHS number. As an NHS number is required to receive a vaccine, and all vaccine records are sent centrally to the IIS, we are confident that when there is no vaccine recorded the person is unvaccinated.

Is it possible less likely to have reported RSV vax status in unvaccinated with GBS?
As the data sources are independent there should not be this bias.

159/160: one of the other central benefits of the SCCS is that it controls for intra-person confounders; assuming that those choosing to get vaccinated might be meaningfully different than those that do not.

We agree this is a benefit. This is now well documented for the methods so do think we need explicitly to mention this. We give a reference for the method.

Results:

182: The COVID-19 pandemic makes this ecologic analysis somewhat challenging. We know that circulation of other important viruses decreased during COVID and other viruses are important causes of GBS. Do you have the possibility of looking at any historical data from before the pandemic? If these data are not available please consider mentioning this in the discussion.

The ecological assessment used data from March 2021 onwards in order to exclude the COVID period where lock down measure were present. The final lockdown finished with schools reopening in March 2021. The figure does show that rates were stable over the period used and we think using just data post the main two pandemic waves is best.

211-215: can you discuss potential reasons for this in the discussion? It would seem that SUS without IVIG would be less severe GBS. Is that the case?

Or is it the case that IVIG was missed or not recorded?

Is this suggesting that the vaccine-associated GBS was more likely to receive IVIG/be more severe? Alternatively, would severe GBS be more likely to have antecedent vaccine noted?

Added to the discussion: SUS GBS cases may include those treated with plasma exchange / plasmapheresis which is considered equivalent efficacy to IVIg therefore use for similar disease severity. However our searches for procedure codes indicating plasma exchange did not return any new information, all patients with a plasma exchange code were also recorded in the IVIg database. It is possible that GBS in SUS patients not appearing in the IVIg database may indicate less severe disease, in line with the clinical criteria for IVIG use but it is important to note that the IVIG and SUS datasets are independently maintained and linked via NHS number. To ensure accurate

determination of vaccination status, all cases included in the analysis were required to link to the population denominator dataset and therefore possess a valid NHS number.

Discussion:

Important to note that the magnitude of the attributable risk per million doses is based on the number of observed/counted cases. In a situation in which this number is inflated (and based on US PPV analyses it is unlikely to be 100%) the absolute magnitude of the AR will always be higher in this type of analysis without chart confirmation. The IRR may be accurate but the absolute value, which is very important in risk-benefit discussions, is likely somewhat of an overestimate. Please consider discussing this further. The supplemental analysis is a nice addition, but of course the PPV is dependent on many things (health systems, knowledge of GBS/RSV association in the local setting, etc).

In the discussion we now note that in general including some-non GBS cases (e.g. PPV is not perfect) does NOT bias the attributable risk UNLESS it is differential by vaccination status (or period post vaccine). This is discussed in paragraph 4 and 5 of the discussion.

Might also note any findings related to maternal program to date in discussion of age-dependent risk (pregnant risk likely to be lower).

We think finding for this group would be a separate paper. As GBS is very rare in this age a longer follow-up of this cohort would be needed.

246: small is relative. Small in absolute terms compared to risk/benefit, but perhaps large in vaccine associated risk terms (eg compared to shingrix, pandemic flu, seasonal flu). Therefore may wish to say a small risk compared to benefit?

In this sentence and abstract "small" is given in the context of comparison to benefit.

259: consider a citation

Citations have been added

261: would be interested to learn more about risk factors considered in risk and control AND unvaccinated patients--were vaccinated and unvaccinated similar? This would be a nice addition if there is space.

Table 1 does include unvaccinated as well as vaccinated by risk and control interval.

We have also responded to the follow up reviewers comments:

Reviewer 2-Follow-up comments:

In the discussion authors have added "Although in US the RSV vaccine is given to a wider age cohort (60 years and over)..." and then discuss the EPIC and FDA studies. Please be aware the FDA study was restricted to adults 65 and older (on Medicaid). – this has been amended in the discussion to make is clear that the FDA study was restricted to adults 65 and older

You note that AR is only impacted by inflated case counts if PPV is differential. I am not sure I understand this because the difference is absolute. Eg 4 cases in risk/2 cases in control leads to an AR of 2 whereas 8 cases in risk/4 cases in control leads to an AR of 4. These both have an IRR of 2 but the absolute attributable risk is different which

means if there is overestimation of both cases and controls (even if overestimation is non-differential) the absolute risk may be different.-

We agree with this point. We were assuming that the PPV as applied to background cases in the vaccine and control periods would be non differential and that all vaccine associated cases would be true cases (by definition). With this assumption then in practice this does actually mean that PPV as measured should be higher in the vaccine period than the control period if there is a true risk (whereas in the US study it was the other way round). This is demonstrated below:

Suppose the vaccine truly causes 2 cases in the risk window and there are 2 true background cases in the risk window (4 total). There are also 2 true background in a similar length control window so RR is 2 and attributable cases 2. Now suppose there is fixed rate of 2 false positives per time period. Then in the risk window we get an extra 2 cases and in the control window an extra 2 cases – so we now have 6 vs 4 cases and a RR of 1.5 but the AR stays at 2 cases. So what we mean is a non-differential rate of false positive cases in the vaccine and control periods. We agree under this scenario that the PPV itself – if all cases were validated would in this case show a higher value in the risk window than the control (4/6 v 2/4).

We now added the following to the discussion

“On the other-hand true differential positive predictive values of background cases in risk window compared to the control window can lead to over estimation of risk, as can attributing a non-GBS vaccine induced adverse event as GBS. Non differential PPV with respect to background cases will not give any bias. Whilst we did not validate cases in our study it is possible differential positive predictive values with respect to background cases were present due to awareness of the hypothesis as GBS was mentioned as an important potential risk following two GBS/Miller-Fisher variant cases in patient information and the Public Assessment Report (20). “